# EGCG-Mediated Protection of Transthyretin Amyloidosis by Stabilizing Transthyretin Tetramers and Disrupting Transthyretin Aggregates

**DOI:** 10.3390/ijms241814146

**Published:** 2023-09-15

**Authors:** Huizhen Zou, Shuangyan Zhou

**Affiliations:** Chongqing Key Laboratory of Big Data for Bio Intelligence, Chongqing University of Posts and Telecommunications, Chongqing 400065, China; s210502021@stu.cqupt.edu.cn

**Keywords:** transthyretin, amyloidosis, EGCG, molecular dynamics simulation

## Abstract

Transthyretin amyloidosis (ATTR) is a progressive and systemic disease caused by the misfolding and amyloid aggregation of transthyretin (TTR). Stabilizing the TTR tetramers and disrupting the formed TTR aggregation are treated as a promising strategy for the treatment of ATTR. Previous studies have reported that epigallocatechin gallate (EGCG) can participate in the whole process of TTR aggregation to prevent ATTR. However, the interaction mechanism of EGCG in this process is still obscure. In this work, we performed molecular dynamics simulations to investigate the interactions between EGCG and TTR tetramers, and between EGCG and TTR aggregates formed by the V30M mutation. The obtained results suggest that EGCG at the binding site of the V30M TTR tetramer can form stable hydrogen bonds with residues in the flexible AB-loop and EF-helix-loop, which reduces the structural mobility of these regions significantly. Additionally, the polyaromatic property of EGCG contributes to the increasement of hydrophobicity at the binding site and thus makes the tetramer difficult to be solvated and dissociated. For V30M-TTR-generated aggregates, EGCG can promote the dissociation of boundary β-strands by destroying key residue interactions of TTR aggregates. Moreover, EGCG is capable of inserting into the side-chain of residues of neighboring β-strands and disrupting the highly structured aggregates. Taken together, this study elucidates the role of EGCG in preventing TTR amyloidosis, which can provide important theoretical support for the future of drug design for ATTR.

## 1. Introduction

Transthyretin (TTR) is a 55 kDa homo-tetrameric protein consisting of 127 residues per monomer in a β-sheet-rich structure [1]. TTR is mainly synthesized and secreted by the liver and is widely distributed in sera and the cerebrospinal fluid (CSF), and the main function of which is to be responsible for the transport of thyroxine and vitamin A–retinol binding protein complexes [2,3,4,5,6]. In the 1970s, the tetramer structure of TTR was first determined by Blake et al. [7,8]. Structurally, the TTR monomer constituting the tetramer contains two β-sheets, termed as the DAGH sheet and the CBEF sheet, and a short α-helix located between the E and F β-strands. Two H β-strands of the adjacent monomers are attached to form a stable dimer, which further forms a tetrameric TTR via a hydrophobic interaction at the dimer interface. Two hydrophobic pockets are contained at the hydrophobic interface of the dimer for the binding of thyroxine (T4) [9].

Normally, TTR exists as a stable tetramer. When mutations take place in the base encoding of TTR, or the structure of TTR is affected by the external environment, the stability of the TTR tetramer decreases, which makes it easy to be dissociated. The dissociated TTR will further misfold and aggregate, triggering transthyretin amyloidosis (ATTR) as a result [10]. ATTR is a progressive and systemic disease that can be classified as being either hereditary (ATTRv) or sporadic (ATTRwt) [11]. ATTRv mainly affects the heart and the peripheral nervous system. It is also a sensorimotor polyneuropathy that causes autonomic dysfunction and gastrointestinal disturbances. Generally, patients with ATTRv will die within 10 years of onset without any treatment [12,13]. Mutations are the main factor giving rise to ATTR, and more than 130 different mutations have been identified to date [14]. Among these identified mutations, V30M is the most common type of ATTRv.

The evident effect of TTR mutations is the reduced stability of TTR tetramers; therefore, the main therapeutic strategies for ATTR at different stages are to reduce or inhibit TTR expression in vivo, stabilize the TTR tetramers, inhibit TTR fibril formation and destroy TTR fibrils. In terms of reducing or inhibiting TTR expression, liver transplantation, gene silencing therapy and gene editing therapy are the main treatments [15,16]. Liver transplantation [17] has been reported to halt disease progression by suppressing the circulation of mutant TTR and a good prognosis has been observed in transplant patients. However, there are still some concerns and long-term complications associated with this treatment strategy since the pre-existing hereditary amyloid deposits may reactivate new circulating TTRwt, promoting amyloid growth and ultimately leading to disease progression [18]. Gene silencing therapy [19] is a strategy for treating ATTR by preventing TTR expression in vivo by silencing the target gene. The use of small interfering RNAs (siRNAs) and antisense nucleotides (ASOs) are the main strategies for the gene silencing of TTR, e.g., Patisiran [20] and Inotersen [21], of which Patisiran was approved for marketing in 2018 [22]. Gene editing therapy [23] is a clustered, regularly spaced, short palindromic repeat-sequence-based Cas9 nucleic acid endonuclease (CRISPR-Cas9) therapy, represented by NTLA-2001 [24], which is currently in the early stages of clinical research. However, these above types of drugs and treatments are extremely expensive and inconvenient. For example, the price of Patisiran is quite expensive and regular injections are required for patients. Additionally, gene editing therapies can lead to unpredictable side effects, such as unexpected changes to other genes, which may lead to other health problems [25].

Compared to liver transplantation, gene silencing and gene editing, small molecule drug treatment has obvious advantages [26]. Firstly, small molecule drugs are convenient to treat, as they are not restricted by time and place. Secondly, they are relatively inexpensive, and have relatively low side effects with a fast metabolism [27,28]. The small molecule treatment of ATTR is mainly achieved by stabilizing the TTR tetramer, since the stability of the TTR tetramer is a decisive factor in regulating the tetramer’s dissociation and is the rate-limiting step in TTR aggregation and fibril formation [29]. Currently, the most widely studied stabilizers for stabilizing TTR tetramers are Diflunisal [30], Tafamidis [31], AG10 [32], Tolcapone [33], Curcumin [34], EGCG [35], etc.

In studies of TTR-related small molecule drugs, it was found that the T4 binding pocket is the main site for most TTR stabilizers. Compared to these T4-binding-site stabilizers, the site for EGCG binding is entirely distinct. Studies have reported that there are three EGCG binding sites in the TTR tetramer structure [36]. The binding sites of EGCG in TTR tetramers and the structure of EGCG are illustrated in Figure 1a,b. Site 1 is sandwiched between monomer A and monomer D and there are two site 1s at two sides of the TTR tetramer. Sites 2 and 3 are located at the same position as monomers A and B, which share the same binding model with EGCG. It has been declared that the EGCG-induced nontoxic oligomer can be formed if EGCG binds at sites 1, 2 and 3 [36]. Moreover, the stability of TTR tetramers can be enhanced if EGCG binds at site 1, acting as a typical TTR tetramer stabilizer. Importantly, studies have reported that the combination of EGCG and other agents can produce synergistic effects, thus further enhancing the stability and anti-amyloidogenic ability of TTR [36,37].

In addition to the effectiveness of EGCG on TTR tetramers, its disruptive effect on formed amyloid fibril aggregates has also been widely reported. It has been reported that EGCG can inhibit the formation of amyloid fibrils by interfering with the self-assembly process of amyloid fibrils and can disrupt the already-formed amyloid fibrils [35,38,39]. A transthyretin-derived amyloid fibril from a patient with hereditary ATTR is shown in Figure 1c. Since proteolysis precedes fibril formation, there are separated N-terminals (residues P11-K35) and C-terminals (residues G57-T123) in the ATTR-associated fibril [40], including three arches from residues P11-K35, K70-L111 and T106-T123.

Considering the specific roles of EGCG in the process of TTR aggregation and amyloid formation, EGCG may be a potential solution for ATTRv treatment. Therefore, we intend to investigate the interaction mechanism of EGCG with TTR tetramers and TTR oligomers via molecular dynamics simulation. Although many studies have identified the stabilizing effect of EGCG on TTR tetramers and the disruption on TTR amyloid fibrils, the current studies are mainly experimental, and information on the interaction mechanism at the molecular level is still limited. Compared to experimental methods, a molecular dynamics simulation can provide transient structural change and the time-evolution structural dynamics of biomacromolecules at the atomic level.

## 2. Results

### 2.1. EGCG Stabilizes the V30M TTR Tetramer

First, to investigate the interaction of EGCG at site 1 of the V30M TTR tetramer, we performed molecular dynamics simulations of the V30M TTR tetramer both with and without EGCG binding at site 1. The “Holo” and “Apo” tetramers were used to denote the systems with and without EGCG binding, respectively. We monitored the conformational stability of the TTR tetramer during the simulation by calculating the RMSD values of the Cα atoms, referenced to the first frame of the trajectory. From Figure 2a, it can be seen that the RMSDs for all systems were less than 2 Å, suggesting that both the Apo and the Holo TTR tetramers are extremely stable even in pathogenetic mutation conditions (V30M). This result explains well that tetramer dissociation is the rate-limiting step for TTR misfolding and aggregation. In addition, it is clear that the RMSD values of the Holo tetramer with EGCG binding were smaller than those of the Apo tetramer, indicating that EGCG binding enhances the stability of the V30M TTR tetramer to some extent.

In addition, since hydrogen bonding at the dimer–dimer interaction interface is an important factor in maintaining the stability of TTR tetramers [41,42], we calculated the number of hydrogen bonds at the dimer interface for each system. There was no significant difference in the dimer interfacial hydrogen number of the Apo and the Holo tetramers, as plotted in Figure 2b. Although EGCG binding at site 1 does not increase the hydrogen bonding interactions between the TTR dimers, hydrogen bonds can be formed between EGCG and residues of site 1 with a total number of about 15. The strong hydrogen bonding interaction between EGCG and residues of site 1 can act as an “anchor” to fasten the four TTR monomers together, making the tetramer bind more tightly.

Subsequently, to quantify the binding affinity of EGCG, we calculated the binding free energy of two EGCGs at site 1 via the method of MM/GBSA. As shown in Table 1, the binding free energies of the two EGCGs are −25.22 kcal/mol and −23.73 kcal/mol, respectively, which indicates that EGCG can bind stably at both site 1s. In comparison, the electrostatic interactions contribute mostly to the EGCG binding, which fits well with the above hydrogen bond analysis.

We further decomposed the binding free energy and calculated the contribution of each amino acid residue at the binding site. As shown in Figure 3a, only those residues with an energy contribution greater than 1 kcal/mol in absolute values were displayed. It can be seen that residues with a large contribution include D18, V20, R21, L82 and I84. As depicted in Figure 3b, D18, V20 and R21 belong to the AB-loop region, and L82 and I84 belong to the EF-helix-loop region. Importantly, it is worth noting that the important role of the above two regions in maintaining TTR tetramers has been previously reported. Ferguson et al. indicated that D18 in the AB-loop plays a critical role in the stabilization of the weak dimer interface [43], and mutants of D18 disrupt the weak interface and render the protein monomeric [44,45]. Our recent study also found that the interaction of residue pair V20-V20 in opposing dimers locks the tetramer together, and the breaking of the “lock” role of V20 is the last step for tetramer dissociation [46]. As for the role of the EF-helix-loop region, previous studies indicate that any change in this region can affect the dimer–dimer interface [43,47], suggesting the critical role of the EF-helix-loop in the structural stability of TTR tetramers.

We infer that the binding of EGCG with residues at the AB-loop and EF-helix-loop should reduce the structural flexibility of these loops. To confirm this, the root mean square fluctuation (RMSF) of residues in the AB-loop and EF-helix-loop regions were calculated for both systems. It can be observed in Figure 4 that the RMSFs of residues in the Holo AB-loop of chain A and chain D were essentially smaller than those of the Apo system. Although, the RMSFs of the residues in chain B and chain C were not reduced, except for residue V20; the difference in the RMSFs of the Apo and Holo systems was relatively small. By contrast, an obvious decrease in the RMSFs was found for most of the EF-helix-loop residues. Therefore, this result verifies our speculation that the binding of EGCG at site 1 of TTR tetramers can indeed reduce structural mobility and thus increase structural stability.

Subsequently, to intuitively visualize the EGCG binding after simulation, we performed a cluster analysis based on the last 50 ns of the trajectories and then extracted a representative conformation with the largest percentage. As presented in Figure 5a, both EGCGs were surrounded by the residues displayed in Figure 3, which ensured a tight binding of the EGCGs. We also performed a hydrophobic interaction and hydrogen bonding analysis based on the representative structure of the cluster analysis to visualize the details of the EGCG binding at the two-dimensional interaction. The LigPlot+ v.2.2 software [48,49] was utilized to generate the two-dimensional interaction networks. As shown in Figure 5b, EGCG1 was found to form hydrogen bonds with the D18 and V20 of chain A, and the S85 of chain D. For EGCG2, hydrogen bonds were formed between EGCG2 and the S85 of chain B, and the D18, V20, R21, S23 and Y78 of chain C. Hence, residues D18, V20 and S85 are common residues to form hydrogen bonds with EGCG at both site 1s, explaining well the large contribution of D18 and V20 in binding with EGCG. The extensive hydrogen bonds formed between the two EGCGs and the residues in the two site 1s can largely attribute to the structural feature of the polyhydroxyl group of EGCGs, which can provide sufficient acceptors and donors for hydrogen bond formation. In addition, the R21, S23, L82 and I84 of chain A, as well as the R21, I84 and P113 of chain D are the main contributors to the hydrophobic interaction with EGCG1. The I84 of chain B, V20, W79, L82, and the I84 of chain C are involved in the hydrophobic interaction with EGCG2. The result indicates that EGCG is able to be surrounded by hydrophobic interactions at the two site 1s of TTR tetramers due to the polyaromatic property of EGCG. The hydrophobic interactions between EGCG and the tetramer give rise to the increasement in the hydrophobicity at the dimer interface, and thus make the solvation more difficult.

Taking the above results together, we can conclude that EGCG binding at site 1 of the tetramer, which is a non-thyroxine binding site, can reduce the flexible AB-loop and EF-helix-loop via a hydrogen bond interaction and hydrophobic interaction. On the one hand, the hydrogen bonds formed between EGCG and the residues at site 1 can fasten the four TTR monomers tightly to enhance stability. On the other hand, the hydrophobic interaction between EGCG and TTR tetramers can also contribute to increasing the difficulty of solvation and thus raise the energy barrier for tetramer dissociation.

### 2.2. Effect of EGCG on the Structure of V30M TTR Aggregates

As described in the Introduction section, EGCG can interact with both the TTR tetramers and the formed aggregates. Based on this, we further performed 200 ns molecular dynamics simulations of TTR aggregates both with and without EGCG to investigate the interactions between EGCG and TTR aggregates. Firstly, we calculated the RMSD values of the Cα atoms of the aggregates, referenced to the first frame of each trajectory. The RMSD plots in Figure 6a suggest that all four systems reached a relatively stable state in the last 50 ns. In comparison, the systems with EGCG exhibited a larger RMSD than the systems without EGCG, especially for the 5-mer system. This result indicates that the binding of EGCG can affect the structure of TTR oligomers.

We then calculated the β-sheet contents of the TTR aggregates for a quantitative characterization of the structural change, since the aggregates are of a β-sheet-rich structure. Here, the β-sheet content is defined as the ratio of the number of residues adopting the β-sheet to the total number of residues. It is revealed in Figure 6b that the addition of EGCG to both the pentamer and decamer can lead to a decrease in β-sheet content. The pentamer in the 5-mer system had a β-sheet content of approximately 50%, while the β-sheet content of the pentamer in the 5-mer + EGCG system dropped to around 45%. The same tendency was observed in the decamer system, although the decrease was smaller. The result of the decreased β-sheet contents suggests that EGCG enables the interruption of the structure of TTR aggregates, which is consistent with the RMSD change depicted in Figure 6a. As with the above β-sheet content analysis, snapshots extracted with an interval of 40 ns also revealed a larger structural perturbation for the systems with EGCG, especially for the pentamer (Appendix A). From Appendix A, it can be seen that the ordered oligomer structures were disrupted by EGCG, compared with the initial pentamer and decamer (0 ns). It is worth noting that the dissociation of the boundary chain was observed in both the 5-mer + EGCG and 10-mer + EGCG systems.

We will then examine the details of the EGCG binding on the aggregates. The plots are displayed in Figure 7, and each β-strand chain is labeled with letters for clarity. Significantly, two approaches were observed for the effects of EGCG on the binding to the pentamer and decamer of the TTR aggregates. As highlighted in the red cycle of Figure 7a,b, the largest influence of EGCG is to promote the dissociation of the boundary β-strands, if the EGCG binds at the edges of the aggregates. In the system of 5-mer + EGCG, EGCG binding at site 1 and site 2 led to the dissociation of β-strands A and E (Figure 7a), respectively. A similar effect was also observed in the system of 10-mer + EGCG with the dissociation of boundary β-strands A and J (Figure 7b). It is worth noting that there are several charged residues at both site 1 and site 2 in Figure 7a,b, such as R34, K35, E62 and E63 in site 2 in Figure 7a, and E61 and E62 in site 1 in Figure 7b. This finding reveals that electrostatic interactions should play an essential role for EGCG binding at the TTR aggregates. Meanwhile, to confirm the dissociation of the boundary β-strands, we also calculated the backbone hydrogen number between the edge β-strands for the pentamer and decamer from the last 50 ns of trajectories, since the β-sheet structure is maintained by the backbone hydrogen bond between the two neighboring β-strands (Appendix A). There was a clear reduction in the backbone hydrogen bond of the boundary β-strands in the 5-mer + EGCG system as compared to that of the 5-mer system. For the decamer systems, the decrease in the backbone hydrogen bond was also observed between the boundary β-strands I and J in the 10-mer + EGCG system, while the reason for no decrease in the hydrogen bonds between boundary A and B can be explained by EGCG being bound at the short N-terminal of the decamer (site 1 in Figure 7b); thus, this may have slightly affected the dissociation of the C-terminal of the β-strand. Importantly, according to the dock-lock mechanism of amyloid aggregation, fibril elongation is closely related to the addition of free monomers to the formed aggregates (template) [50,51]. Therefore, the dissociation of the boundary β-strands from the aggregates should serve as a vital process for oligomer disruption, and recent studies reported by Liu et al. also emphasized the significant role of the dissociation of boundary chains from formed Tau fibrils [52,53].

The second approach of EGCG for disrupting TTR aggregates is that EGCG can insert into two neighboring β-strands, as highlighted in the blue cycle labeled in Figure 7a,b. In Figure 7a, one of the aromatic rings of the EGCG at site 3 is entirely embedded in the side chains of R21(B) and R21(D) of the pentamer. Similarly, the two EGCG molecules at site 3 in Figure 7b were also found to insert into the side chains of R21(I) and R21(H), and two hydrogen bonds were formed between the EGCG, colored in green, and the V20(I) and V20(H). In addition, there were three EGCG molecules gathered in site 4 of Figure 7b, which also led to the position displacement of the side chain of M13. Therefore, the insertion of EGCG into the adjacent β-strands can certainly trigger the rearrangement of the side chains of surrounding residues, which will further destroy the highly ordered structure of the aggregates.

Finally, to elucidate how EGCG disrupts the TTR aggregates, we monitored the internal interaction within the aggregates, and the interactions between EGCG and TTR aggregates. Since the pronounced effect of EGCG observed in disrupting the TTR aggregates is the dissociation of boundary chains in system 5-mer + EGCG, we therefore focused on the interactions at site 1 and site 2, as seen in Figure 7a. As the residue–residue contact map of the boundary chains, illustrated in Figure 8a, the inter-chain interaction of the TTR aggregates at the EGCG binding sites changed. Compared to the 5-mer system, the inter-chain interaction decreased in both sites of system 5-mer + EGCG. At binding site 1, there were two obvious differences, corresponding to the interaction between fragment 98–104 of chain A and fragment 97–104 of chain B, and between fragment 100–108 of chain A and fragment 74–77 of chain B. At binding site 2, the inter-chain interaction between fragment 56–65 of chain D and fragment 56–65 of chain E dropped significantly. The decreased inter-chain interaction in the 5-mer + EGCG system reveals the dissociation tendency of the boundary chains at these regions.

In addition, as shown in Figure 8b, the calculation of the hydrogen bond occupancy between EGCG and the residues in site 1 and site 2 suggests that charged residues benefit the binding of EGCG at these two sites, including R34, K35, E61, E62, E63, R103 and R104. Among these charged residues, K35, E63 and R103 play an essential role in maintaining the topology of TTR aggregates. From the experimental model (crystal structure) depicted in Appendix A, it can be found that the K35 in each β-strand chain forms a salt bridge with E63 in the same chain, which ensures a tight binding between the N-terminal and C-terminal of the aggregates. R103 is capable of forming a stable salt bridge with D99 and D74 in the same chain, and these salt bridges are of vital importance to keep the arch topology of TTR aggregates. However, these aforementioned salt bridges are partially destroyed upon the binding of EGCG, e.g., the salt bridges R103-D74 of chain A and K35-E63 of chain E, as shown in Appendix A and S3c. Once the salt bridges are destroyed by EGCG, it will be difficult to maintain the original residue interactions and thus promote dissociation.

## 3. Discussion

EGCG is the major component of green tea polyphenol and has been extensively reported to be a potential compound that can be used in the treatment of amyloidosis, e.g., Alzheimer’s disease [54] and Huntington’s disease [55]. ATTR is a form of systemic amyloidosis resulting from the accumulation of TTR amyloid fibrils in various organs and tissues [56]. Focusing on the effects of EGCG on ATTR, we studied the interaction between EGCG and TTR tetramers, and between EGCG and TTR aggregates. EGCG binding at site 1 of TTR tetramers increases the structural stability by reducing the flexibility of the AB-loop and the EF-helix-loop. The above loops were previously reported to be the critical regions for the misfolding and aggregation of TTR [43,57]. The key residues involving EGCG binding at site 1 include D18, V20, R21, L82 and I84, as revealed by the decomposition of the binding free energy, as shown in Figure 3. Importantly, it is reported that the mutation of D18 to alanine can prevent the inhibitory effect of EGCG [36,47], which also suggests the crucial role of D18 in EGCG binding. Moreover, due to the polyhydroxyl and polyaromatic properties, there are extensive hydrogen bond interactions and hydrophobic interactions between EGCG and residues around site 1 (Figure 5), ensuring the tight binding of EGCG. On the one hand, the hydrogen bond interaction between EGCG and the residues in site 1 can fasten the four monomers tightly and reduce the flexibility of the loops, as we declared above. On the other hand, the binding of EGCG can fill the cavity of site 1 and thus increase the interfacial hydrophobicity, making the tetramer difficult to be solvated and dissociated.

Regarding the effect of EGCG on disrupting the TTR aggregates, it can promote the dissociation of boundary chains if EGCG binds at the edge of aggregates, such as in site 1 and site 2, as depicted in Figure 7a,b. In addition, as in the blue labeled sites shown in Figure 7, EGCG can also trigger the rearrangement of the sidechains of residues by inserting into the residues of neighboring chains. It is worth noting that charged residues can be found in all the labeled sites in Figure 7, including R21, K35, R34, E61, E62, E63, K80, R103 and R104. By examining the crystal structure of TTR aggregates, we found that some of these residues play a significant role in maintaining the topology of the aggregates by forming salt bridges with related residues (Appendix A), while the binding of EGCG with these residues can certainly destroy the native residue interactions, as shown in Appendix A. Recent studies also reported the similar effect of EGCG on disrupting the key K28-A42 salt bridge of the Aβ42 oligomer, leading to the destruction of the Aβ42 oligomer [58,59].

Taken together, we unraveled the EGCG-mediated protection of TTR amyloidosis. Despite the conclusion of this work being drawn based on the structural models of V30M, the observation should also be applicable to WT and some other variants of TTR. As shown in Appendix A, residue 30 in the TTR tetramer is far from binding site 1, which means that the substitution of residue 30 should have little effect on the binding of EGCG to site 1. Meanwhile, a recent study reported by Leach et al. found that the changes in the ^13^Cα chemical shift provoked by V30M mutation are observed for V16, T49, I73 and A109 [60], none of which are in binding site 1 of TTR tetramers. Therefore, EGCG should share the same binding between WT and V30M. In addition, the structural superposition of WT-TTR and V30M TTR aggregates suggests a high structural similarity, except for a tiny difference near residue 30 (Appendix A). More importantly, previous biological experiments uncovered that EGCG can certainly inhibit the TTR amyloid fibril formation of wild-type TTR [61] and other common TTR mutants, e.g., L52P and Y78F [39,62]. Based on the above discussion, it can be speculated that the findings of this work should also be applicable to WT or some other variants of TTR.

## 4. Materials and Methods

### 4.1. Structural Preparation

The structure of the EGCG-V30M TTR tetramer complex was obtained from the protein data bank and the PDB ID was 3NG5 [36]. The structure was obtained via X-ray diffraction with a resolution of 1.7 Å. Since the main purpose of this work is to focus on the stabilizing effect of EGCG on the TTR tetramer, we only kept two EGCGs at site 1. The crystal structure obtained from the PBD contains only the coordinates of the TTR dimer with EGCG; therefore, the transformation matrices in the PDB files were used to generate the complex structure of the TTR tetramer and two EGCGs at site 1 (Appendix A).

In the system for studying the effect of EGCG on TTR aggregates, the structure of fibrils from a patient with hereditary ATTR was used (PDB ID: 6SDZ) [40]; the structures were determined via the method of cryo-electron microscopy with a resolution of 2.97 Å. In this work, a pentamer and a decamer were extracted from the V30M ATTR-associated amyloid fibril to represent the TTR aggregates. Here, the TTR pentamer and decamer were used for simulation, with the purpose of exploring the similarities of the different aggregates. Seven and fifteen EGCG small molecules were randomly placed around the TTR pentamer and decamer to construct the simulated system, referring to the work of Wang et al. [63] (Appendix A). We denoted these systems as 5-mer, 5-mer + EGCG, 10-mer and 10-mer + EGCG. A structural optimization of EGCG was first performed using the Gaussian 09 software, with the B3LYP method and 6-31++G basis set. The partial atomic charges were assigned using the restrained electrostatic potential (RESP) approach [64] and the general AMBER forcefield (GAFF) [65] was used to describe the EGCG.

### 4.2. Molecular Dynamics Simulation

All molecular dynamics (MD) simulations [66] were performed by using the AMBER 18 program [67], and the AMBER FF14SB force field [68] was used to describe the proteins. All systems were performed in an explicit solvent model with periodic boundary conditions [69]. The TIP3P solvent model [70] was used to describe the solvent effect by adding a solvent box with a thickness of 12 Å from the edge of the solute, and the solvent box was filled with Na^+^ to keep the system neutral [71]. Prior to the dynamics simulations, each system was energy minimized in three steps: first, by keeping the protein and EGCG constraint and minimizing the water molecules; second, by keeping only the protein backbone bound and minimizing the side chains; finally, by keeping all molecules free. After energy minimization, each system was subjected to a 1 ns equilibrium, making the system parameters basically constant. Finally, a total of 200 ns of MD simulations were performed for the three systems at 310 K and 1 atm isothermal isobaric (NPT), respectively, and the obtained atomic coordinates were saved every 10 ps for analysis. As control systems, the V30M TTR tetramer and V30M ATTR-associated aggregates (without EGCG) were also simulated, using similar procedures as described above. During the molecular dynamic simulations, the temperature was controlled using Langevin kinetics and the Langevin damping factor was set to 5 ps. The pressure was kept constant using the Langevin piston Nosé–Hoover method [72]. Electrostatic interactions were calculated using the particle grid Ewald method [73] and the van der Waals interactions were smoothly closed at intervals of 10 to 12 Å. Using the SHAKE algorithm [74], all bonds involving hydrogen atoms were considered rigid, allowing integration time steps of 2 fs.

### 4.3. Calculation of Binding Free Energy

The MM-GBSA (molecular mechanics–generalized Born surface area) [75] method was used to calculate the binding free energy of EGCG with the V30M TTR tetramer and V30M ATTR-associated oligomer, respectively. Five hundred snapshots were extracted from the last 100 ns of the simulation, i.e., 100 to 200 ns and 500 to 600 ns, respectively, with a time interval of 200 ps. For each snapshot, we calculated the binding free energy (Δ*G_bind_*) as follows:
Δ*G_bind_* = *G_complex_* − (*G_receptor_* + *G_ligand_*)(1)
Δ*G_bind_* = Δ*E_MM_* + Δ*G_sol_* + *T*Δ*S*(2)
*E_MM_* = *E_vdw_* + *E_ele_* + *E_int_*(3)
*G_sol_* = *G_GB_* + *G_SA_*(4)
where *G_complex_*, *G_receptor_* and *G_ligand_* represent the free energy of the complex of EGCG and TTR, the V30M TTR, and the EGCG small molecules, respectively. *E_MM_* is the total gas energy calculated using the molecular mechanics force field of AMBER ff14SB, which includes the van der Waals forces *E_vdw_*, electrostatic forces *E_ele_* and internal energy *E_int_* generated from bonds, angles and dihedrals. *G_sol_* is the solvation free energy, which can be decomposed into nonpolar (*G_GB_*) and polar (*G_SA_*) components. The solvent accessible surface area (SASA) was estimated by using the MSMS algorithm [76] with a probe radius of 1.4 Å. The polar part was calculated using the generalized Born (GB) method with the igb set to 2; the salt concentration (Saltcon) was set to 0.1 [77]. Here, we omitted the effect of the conformational entropy for two reasons. Firstly, the binding free energy calculation was performed for the purpose of identifying key residues for EGCG binding. Therefore, a qualitative evaluation is sufficient. Secondly, entropy calculation is time-consuming and computationally intensive. If a small number of snapshots are used, the calculated entropy will fluctuate greatly [78].

## 5. Conclusions

In this study, we investigated the effect of EGCG on the TTR tetramer of V30M and on V30M-generated aggregates via molecular dynamics simulations. Due to the polyhydroxyl groups of EGCG, the binding of EGCG at the two site 1s of TTR tetramers can form extensive hydrogen bonds with residues of the AB-loop and the EF-helix-loop regions, which increases the structural stability of these amyloid-related regions and stabilizes the TTR tetramer as a result. These key residues identified via the decomposition of the binding free energy are D18, V20, R21, L82 and I84. In addition, there are also three aromatic rings in the structure of EGCG. This property makes it possible to form a hydrophobic interaction between EGCG and the residues at site 1 of the TTR tetramer, which can also increase the hydrophobicity at the dimer interfacial and thus make it difficult to be solvated at the dimer interface. Therefore, both the hydrogen bond formation and hydrophobic interaction are the main factors for the stabilization effect of EGCG on TTR tetramers, which can largely attribute to the polyhydroxyl groups and aromatic rings in EGCG. Further investigation of the influence of EGCG on TTR aggregates suggests that EGCG should disrupt the TTR aggregates via two approaches. The first and most important effects is to promote the dissociation of boundary β-strands by destroying key residue interactions, if EGCG binds at the edge of aggregates. In addition, EGCG can also insert into the orderly arranged side chains of residues, which will trigger the structural rearrangement and further destroy the structural morphology of the aggregates.

Overall, this work reveals both the protective mechanism by which EGCG protects V30M TTR tetramers from dissociation and the mechanism by which EGCG disrupts the already-generated aggregates. The result of this work can contribute to the development of new drugs with similar mechanisms that act to inhibit multiple stages of protein amyloidosis.

## Figures and Tables

**Figure 1 ijms-24-14146-f001:**
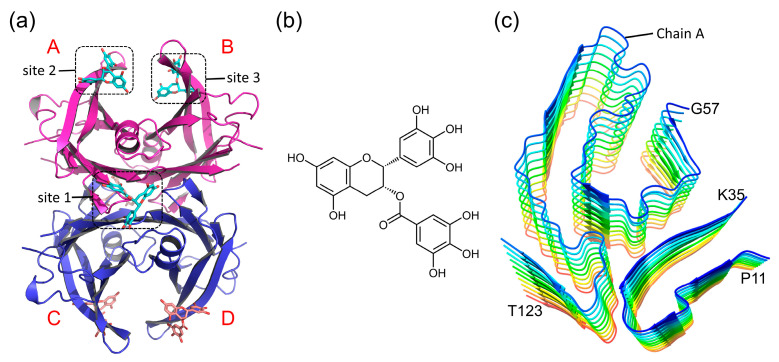
Molecular structure. (**a**) The tetrameric structure of V30M TTR. The monomeric subunits are labeled with letters A–D; (**b**) EGCG; (**c**) TTR amyloid fibril.

**Figure 2 ijms-24-14146-f002:**
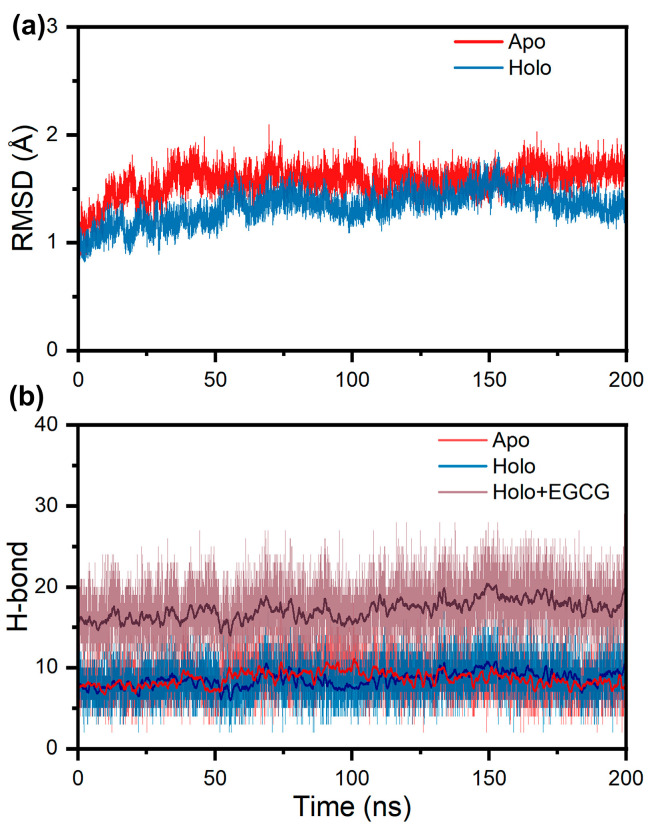
Structural characterization of TTR tetramer during simulation. (**a**) RMSD plots of Apo TTR tetramer and Holo tetramer with EGCG binding; (**b**) plots of the dimer interfacial hydrogen bonds of Apo tetramer and Holo tetramer as well as the dimeric interfacial hydrogen bonds for Holo tetramer, plus hydrogen bonds formed between EGCG and Holo tetramer.

**Figure 3 ijms-24-14146-f003:**
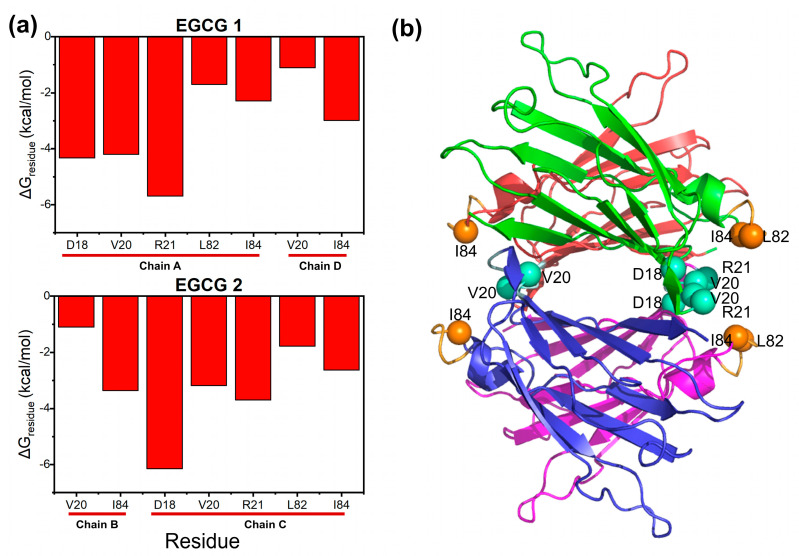
Residues that contribute to the interaction of EGCG. (**a**) Residue free energy decomposition of EGCG interacting with V30M TTR; only residues with energy contribution larger than 1 kcal/mol were displayed; (**b**) the location of large contribution residues in the tetramer structure. Residues were represented as balls; the sky-blue balls are residues in the AB-loop and orange balls are residues in the EF-helix-loop.

**Figure 4 ijms-24-14146-f004:**
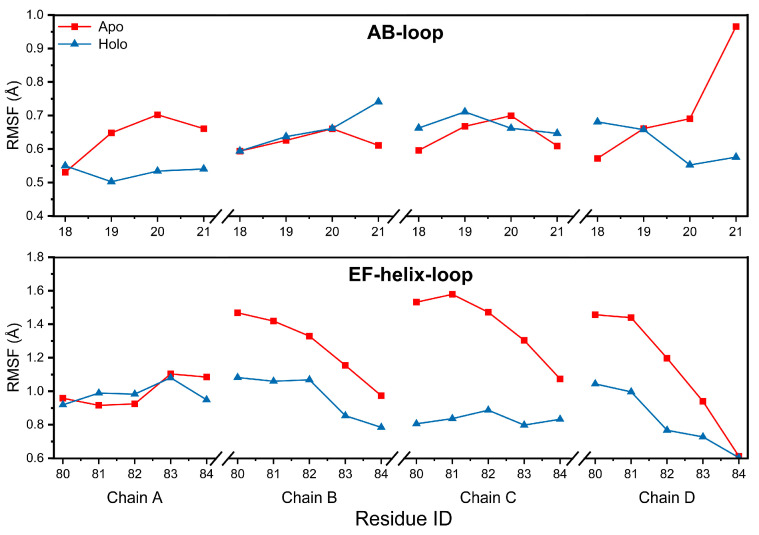
RMSF values of residues in the AB-loop and EF-helix-loop regions for the Apo and Holo TTR tetramers.

**Figure 5 ijms-24-14146-f005:**
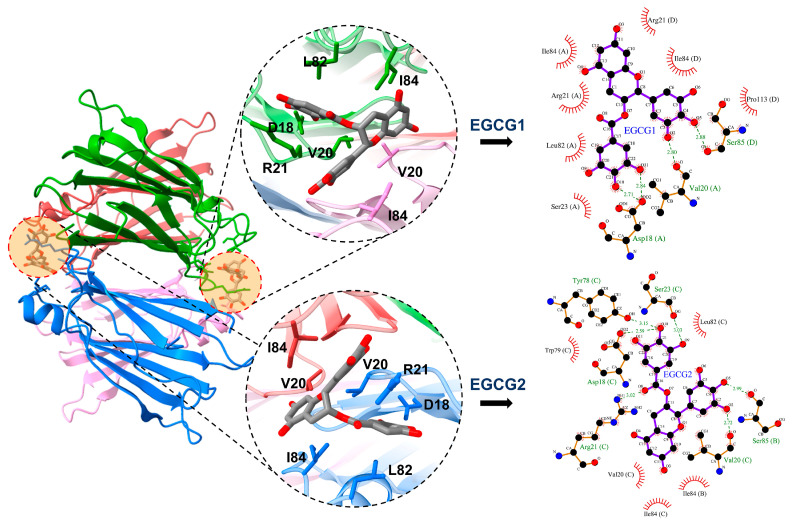
Representative structure of EGCG binding at TTR tetramer and corresponding hydrophobic and hydrogen bonding interaction of EGCG with V30M TTR at binding site 1. Letters A–D represent for corresponding chain of tetramer.

**Figure 6 ijms-24-14146-f006:**
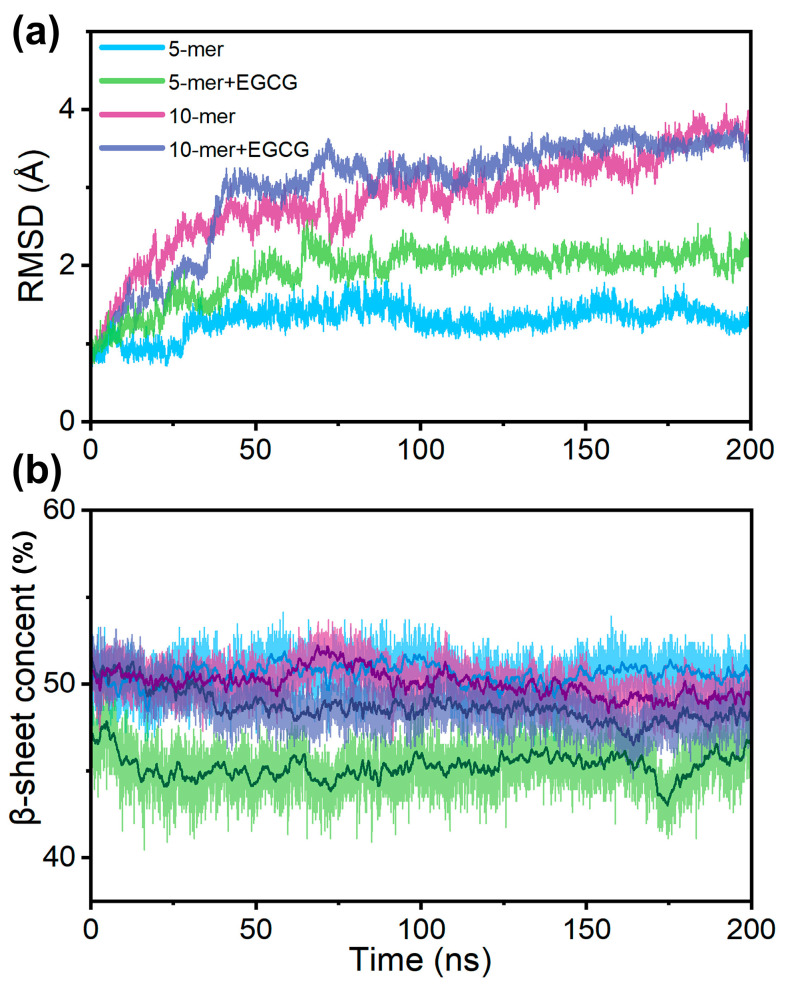
Structural change in TTR aggregates evaluated using (**a**) the time evolution of RMSDs of Cα atoms, referenced to the first structure of each trajectory, and (**b**) the change in β-sheet content during the simulation for each system.

**Figure 7 ijms-24-14146-f007:**
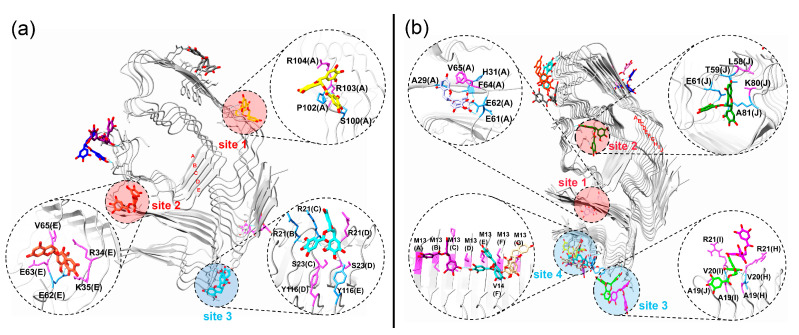
Representative structures of EGCG binding with pentamer and decamer of TTR. (**a**) Interaction of EGCG with pentamer of TTR; (**b**) interaction of EGCG with decamer of TTR. The red and blue cycles refer to two methods by which EGCG disrupt TTR aggregation. Letters in the picture represent for the corresponding β-strand chains of pentamer and decamer.

**Figure 8 ijms-24-14146-f008:**
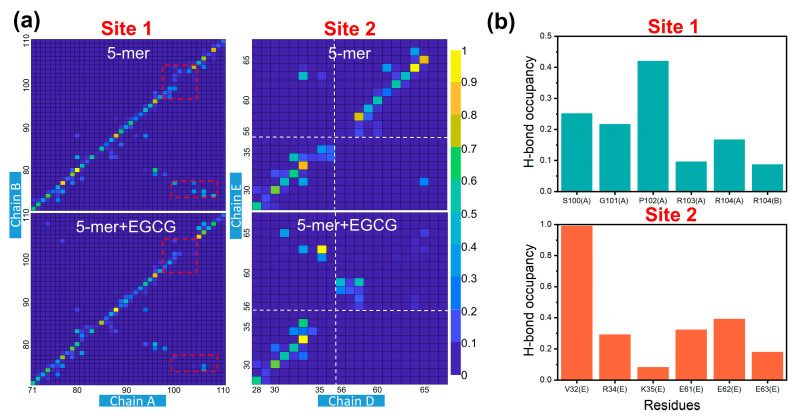
Calculation of interactions at site 1 and site 2. (**a**) Residue–residue contact maps for boundary chains. The significant differences between 5-mer and 5-mer + EGCG are highlighted by dashed red boxes; (**b**) hydrogen bond occupancy analysis between EGCG and residues in site 1 and site 2.

**Table 1 ijms-24-14146-t001:** Binding free energy calculation for EGCG binding at site 1 of TTR tetramer.

	EGCG1 (kcal/mol)	EGCG2 (kcal/mol)
Δ*E_ele_*	−64.18	−61.18
Δ*E_vdw_*	−34.92	−34.77
Δ*E_int_*	0.00	0.00
Δ*E_MM_*	−99.10	−95.95
Δ*G_GB_*	79.60	77.95
Δ*G_SA_*	−5.72	−5.73
Δ*G_sol_*	73.88	72.22
Δ*G_bind_*	−25.22	−23.73

## Data Availability

All initial structures and input files for simulation were deposited at GitHub, and the URL is https://github.com/zouhz2023/initial-structures.git (accessed on 5 August 2023).

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
