# Peer review of "EGCG-Mediated Protection of Transthyretin Amyloidosis by Stabilizing Transthyretin Tetramers and Disrupting Transthyretin Aggregates"

_ijms, 2023, doi:10.3390/ijms241814146_

Round 1
Reviewer 1 Report
This manuscript reported the computational study to appreciate the detailed interaction mechanism of EGCG with TTR in its native and aggregated states. I think that this study provided several useful information, such as the important residues of native TTR for the interaction with EGCG and the placeable explanation how EGCG disrupts TTR aggregates, which may help to develop novel therapeutic molecules for TTR amyloidosis. However, I also have a few major concerns that need to be considered before the publication of the manuscript:
- For the interaction between native TTR and EGCG, the authors only considered the ‘site 1’, while the X-ray crystallographic model reported two additional sites. I think that the authors should analyze the effects of EGCGs at the sites 2 and 3, because it is possible that the inhibitory effects of EGCG against TTR aggregation may also partially come from those. Or, it would be necessary for the authors to provide reasonable explanation why they think that EGCG at the sites 2 and 3 does not contribute to the tetrameric stability of TTR.
- In this manuscript, the structural models for TTR(V30M) were employed. I think that the authors should include some discussions whether they think that all the observations made here are also applicable to WT or other variants of TTR.
- The analysis done with TTR aggregates showed the binding of EGCG onto TTR aggregates, but it did not provide direct evidence explaining how the observed interaction with EGCG compromises the stability of TTR aggregates. I think that the observed results rather imply that EGCG may contribute to the overall stability of TTR aggregates by compensating some unfavorable interactions between TTR residues. Could the authors provide additional analysis results which directly demonstrates the disruption mechanism of TTR aggregates?
- It would be helpful to readers if the authors can provide the figure in which the structural model of TTR aggregates (the experimental model) is directly compared with the computational model of EGCG-bound TTR aggregates.
Author Response
Dear reviewer,
Thank you for your careful reading and valuable comments and constructive suggestions, which have significantly improved our manuscript.
We have carefully considered all the comments from the reviewers and made corresponding modifications to our manuscript. The response is uploaded in the attachment.

Reviewer 2 Report
In this current study, the authors conducted molecular dynamics (MD) simulations involving TTR proteins and EGCG molecules. While the research is well-explained and the results are presented effectively, a notable point of consideration lies in the reproducibility aspect of the work. Given the inherently stochastic nature of MD simulations, it is generally advisable to carry out simulations in triplicate or more to ensure the robustness of the obtained data. In this study, however, the authors chose to execute single simulations for each system. To enhance the credibility of the findings, I would suggest the incorporation of additional simulations to bolster the confidence of readers in the presented data.
Author Response
Dear reviewer,
Thank you for your careful reading and valuable comments and constructive suggestions, which have significantly improved our manuscript. We have carefully considered all the comments from the reviewers and made corresponding modifications to our manuscript.
In this current study, the authors conducted molecular dynamics (MD) simulations involving TTR proteins and EGCG molecules. While the research is well-explained and the results are presented effectively, a notable point of consideration lies in the reproducibility aspect of the work. Given the inherently stochastic nature of MD simulations, it is generally advisable to carry out simulations in triplicate or more to ensure the robustness of the obtained data. In this study, however, the authors chose to execute single simulations for each system. To enhance the credibility of the findings, I would suggest the incorporation of additional simulations to bolster the confidence of readers in the presented data.
R:Thank you very much for the valuable comment. We agree the reviewer very well and it should be more reasonable to run parallel trajectory to avoid the randomness of obtained results. Therefore, to ensure the reliability of the results in this work, we compared our results extensively with the existing results. For the system of EGCG and TTR tetramer, we illustrated that EGCG binding at site 1 can enhance the structural stability by reducing the flexibility of AB-loop (L17-P24) and EF-helix-loop (D74-E89). Key residues contribute to the binding of EGCG are D18, V20, R21, L82 and I84. First, the role of the AB-loop and EF-helix-loop in maintaining the stability of TTR tetramer has been previously reported. Ferguson et al. indicated that D18 in the AB-loop plays a critical role in stabilization of the weak dimer interface, and mutants of D18 disrupt the weak interface and render the protein monomeric (Amyloid. 2004, 11, 61-66; Biochemistry 2003, 42, 6656-6663). Our recent study also found that the interaction of residue pair V20-V20 in opposing dimer locks the tetramer together and breaking of the “lock” role of V20 in the AB-loop is the last step for tetramer dissociation (J. Chem. Inf. Model. 2022, 62, 6667-6678). Moreover, the alanine mutation scanning experiment in the cell culture system reveal that the effect of EGCG can be inhibited by the mutation of D18 to alanine (Biochemistry, 2010, 49, 6104–6114). As for the role of EF-helix-loop region, previous studies indicate that any change in this region can affect the dimer-dimer interface (Biochemistry 2021, 60, 756-764; Curr. Med Chem. 2012, 19, 2324-2342.), suggesting the critical role of EF-helix-loop in structural stability of TTR tetramer. Consequently, the conclusion that EGCG binding at site 1 can reduce the flexibility of AB-loop and EF-helix-loop to increase the stability of TTR tetramer should be reasonable.
In the EGCG-TTR oligomer system, four systems were designed, including pentamer and decamer of TTR with or without EGCG. The purpose of designing pentamer and decamer is to acquire the similar trend of EGCG on pentamer and decamer, which resembles to perform a parallel run. Based on the interaction between EGCG and pentamer, we drawn the conclusion that EGCG can promote the dissociation of boundary chain or compel the rearrangement of residue sidechain depending on the binding site. That is, the above findings are observed in both the pentamer and decamer of TTR, which ensures the reliability of our conclusion to some extent.